# Christianity and Fraternal Globalisation

**Darlan Aurélio de Aviz** [1] and **Luís M. Figueiredo Rodrigues** [2,*]

1  Faculdade de São Bento, Rio de Janeiro 20090-030, Brazil; d.aviz@professor.fsbrj.edu.br
2  Faculdade de Teologia, Universidade Católica Portuguesa, 4710-362 Braga, Portugal
*  Correspondence: figueiredorodrigues@ucp.pt

**Abstract:** This paper discusses the importance of fraternity in building more fair and humane societies. The globalised world poses new challenges to the understanding of fraternity, especially when it is seen as an attack on the dignity of each person. Religions are considered to have an important role to play in responding to the dehumanisation that a certain type of globalisation can promote. To this end, we reflect on the threats and opportunities that the globalised world offers and how religious thought can make a positive contribution. Based on Pope Francis' challenges, we will respond to what the "something greater" is that fraternity has to offer the world. The answer is developed using sources from the sacred scripture and the patristics to understand Christianity, where it is concluded that the model of relationships between people, in the thought of the magisterium in the light of sacred scripture, aims to awaken the awareness of a social friendship, which will grow to the extent that one offers what is best for the other, simply by loving them and being who they are.

**Keywords:** fraternity; globalisation; complexity; Pope Francis; identity





## 1. Introduction

The UN's 2030 Agenda for Sustainable Development Goals (SDGs) calls for the adoption of a public agenda for a collective political and social endeavour aimed at shared responsibility, supported by a culture of solidarity and universal fraternity. At times like these, Christianity can have great potential for developing empathy and "to ensure that all human beings can enjoy prosperous and fulfilling lives and that economic, social and technological progress occurs in harmony with nature" (ONU 2023). It is important for it to be a referential and historical source, valid for building consensus on common life, especially in a globalized context.

Globalisation has been perceived as a process of social interconnection, extending across the globe. At times, globalisation is seen as a threat to individuals and societies. Therefore, it is important to understand the deeper implications of the concept of "fraternity" and its associated terms, such as "friendship" and "closeness", in an increasingly complex and global world.

For his part, Pope Francis, especially in his encyclical *Fratelli Tutti*, has been warning about the impotence of fraternity for each individual and for each community. While the trilogy of liberty, equality and fraternity of the French Revolution is usually seen as the one that gave fraternity political and social dignity, a closer look reveals that its roots are much older and that it is deeply rooted within each human being. On the other hand, fraternity necessarily requires something greater, which is capable of strengthening and consolidating freedom and equality.

This article seeks to understand what this "something greater" is that Pope Francis invites us to discover and value, in the certainty that fraternity builds not only the Church, but also civil society, which integrates the Church and to which she seeks to offer her specific contribution.

## 2. Contextualization

Globalisation is a complex and multifaceted phenomenon, which technological development has greatly enhanced (McNeill and McNeill 2004). This concept intertwines several dimensions, such as economic, social, political, cultural and religious, which are shaped by the relationship between the various actors and forces involved (Steger 2013). Globalisation, as a phenomenon, has been understood as a process of social interconnection, extending across the globe (Giddens 2006). Interconnection on a global scale—and according to different periods of history and latitudes—has allowed the exchange of cultural meanings between different societies, on the one hand, and the universalization of some concepts and ideas on the other. Therefore, globalisation is perceived as a subjective practice, which interconnects the global whole (Morin 2015; Ritzer 2007). Thus, globalisation may simply be the name given to a matrix of processes that extend social relations across the world space. However, how people experience these relations is quite complex, changeable and difficult to delineate. The relevance of a "global" approach to understanding the world lies in the fact that it is necessary to pay attention to the cultural phenomena of globalisation in order to understand particular events in the light of globalisation, but also the other way round. There are many phenomena which, while now global, were born in a particular social context (Abrantes and Lechner 2022).

Although there are already some studies on the impact of globalisation on ecclesial life and, above all, on the new challenges facing leadership (Ormerod 2007; Gunter 2018), it appears that the religious component of the human being has not been considered when theorising about what it is to lead in a globalised world, even though its relevance has been noted (Alkhatib and Arnout 2019). There is no consensus on the etymological origin of the word "religion", but there are two hypotheses that appear to be the most plausible and that, at the same time, allow a comprehensive approach to it. On the one hand, and from the perspective of Marco Tulio Cicero (106–43 BC), the origin comes from re-legere (re-read), which would mean taking a new perspective, rereading reality from a new principle and, consequently, advocating new options. Thus understood, to speak of religion would be to address a reality that designates a certain way of thinking, acting, and expressing oneself. On the other hand, Lucius Celios Firmianus Lactantius (240–320 AD) suggests that "religion" derives from re-ligare (to connect). A Western view, clearly marked by Greek philosophy and the Abrahamic religions, understands religion as the bond of piety that unites God with humanity, to which the believer responds through obedience. This is understood as more than submitting to something; it is knowing how to listen that generates, as a response, attitudes and behaviours.

That is why in a globalised, hyperconnected world, the myths that are disseminated and that have a clear adherence in the lives of citizens highlight what they want, the narratives in which they want to participate. This is regardless of where they live. It is in this context that Raphaël Liogier's reflection offers a relevant proposal for thinking about the impact of religions in a globalised world (Liogier 2012; Ramón Solans 2019) where technologies act as a facilitator (Liogier 2020) role, promoting narratives of peace as one of the happiest consequences of human fraternity (Liogier 2018). In *Souci de Soi, Conscience Du Monde. Vers Une Religion Globale?* Liogier describes a "mythological metamorphosis" at the planetary level, which he calls "individualism-globalism". His argument builds on the concepts of "myth", "metamorphosis" and "desire to be" to justify that, far from disappearing, the religious person is recomposed in "individualism-globalism". This concept is described as

> the essential and irreducible mythical tension, the mythological heart from which the culture of advanced industrial societies feeds, drawing from it the ethics for sustainable development, as well as the aspiration for personal development, the desire for distant adventure, humanitarian action, but also, correlatively, the balance on the same imaginary tightrope, the search for self-knowledge, and for individual well-being[1]. In short, the construction of identities that articulate the individual and the global.

Since every identity has an irreducibly mythical expression, the mutation of identity produced by globalisation necessarily concerns the religious field. It is for this reason that religion appears to be a crucial and privileged field of research for understanding the complex development of global identities. These are neither uniform nor universal. On the contrary, they are increasingly diverse, because classical, territorialized, and inter-territorialize identities have not disappeared. On the other hand, global, deterritorialized identities do not exist in the abstract, but constitute complex and mutable sets. They reflect, in groups and individuals, the multiple layers of interconnected identities: from territorial to interterritorial and deterritorialized identities. It could therefore be said that a characteristic of global identities is their meaningful multiplicity, even at a micro-local level.

It is in this context that Amartya Sen's proposal has particular significance. He advocates that identity is a social construction and that there is no single way of being human, so the diversity of identities is a richness. On the contrary, violence is the result of the attempt to impose a single identity on everyone. Globalisation can play a dual role here: on the one hand, globalisation can increase violence, as it can lead to greater cultural homogenisation; on the other hand, globalisation can be a catalyst for peace, as it can promote dialogue and understanding between different cultures. This is why Sen proposes a series of measures to promote peace and tolerance in a globalised world. These are *education*, *dialogue* and *cooperation* (Sen 2006; Hamilton 2019).

However, it is undeniable that populist and anti-globalisation sentiments have been growing, at least in some regions of the globe. According to the thinking produced by Ramon Flecha, we can state that people are increasingly isolated, living in their own worlds. The media plays an important role here because, on the one hand, it foments individualisation and, on the other, it gives more space to narratives of violence and destruction, which means that strong relationships based on empathy are not created. On the contrary, protagonists who have justified violence or the lack of truth and authenticity criteria as referents of thought are more emphasised. These factors are further enhanced by the absence of dialogue, as people are not talking and listening as one would wish, with the consequent difficulty in resolving conflicts and building consensus (Flecha 2022).

Arne Bigsten (Bigsten 2020), in the political field, theorises a model for responding to this situation by articulating the three values of the French Revolution: liberty, equality and fraternity. This triad is perceived as the synthesis of what policies must guarantee in order to be of effective service to citizens, providing them with the desired well-being. Here, while freedom and equality have been the object of protection efforts, the same does not apply so much to fraternity. Fraternity has been threatened by globalisation and technological development (Soler Gil 2013).

## 3. Political Motivations

Fraternity has economic, political and social implications of identity, which determine social cohesion. The economic determinant of the threat to fraternity or shared identity is economic inequality, which is reflected in the types of jobs and incomes that can be accessed. The political component concerns national or regional identity. The social component of identity is a sense of shared cultural values (Spicker 2006). As identities become polarised, trust in those in leadership positions begins to fail. Recognition of their significant and even inspirational role as a role model declines. This reduces the ability to recognize their significant and even inspiring role as a role model (Scheler 2018). As a shared identity weakens, solidarity also diminishes, and distrust between social classes grows instead, especially between the least favoured and the most favoured. This is the favourable ground for populisms to spread their narratives, legitimising discourses of hatred and violence, even using the resources of religions.

Therefore, it is essential to understand what factors can combat the phenomena of dehumanisation and violence, with the aim of rapprochement and dialogue between cultures, promoting a culture of peace, rejecting violence and hatred, and realising the contribution of dialogue between different cultures. All this, starting from the realisation that religions

(Francis 2020) raise awareness of the common values of all human beings and create an enabling environment for achieving peace and understanding among all and at all levels: local, national, regional, and global (Delors et al. 1996).

The concept of fraternity, as a desire and as a political realisation, appears in legal texts of states at the time of the French Revolution (1789). More concretely, it was in the Second Constitution of the French Republic (1848) that the term began to appear in legislative texts (Constitution de 1848, IIe République 2022). In addition, the concept of *fraternity* brings together several dimensions in a very specific way: as a kind of mutual aid, cooperation or community; and as a bond at the family, labour or political level. In the face of the enemy, human misery or loneliness, fraternity is perhaps best labelled using the synonym "solidarity". In the face of the foreigner, the stranger or the one who is different for whatever reason, it is called "hospitality" (Moyaert 2011). What is important to emphasise is that fraternity is characterised by the impossibility of being imposed by the force of those in authority. It must arise from each individual, since fraternity is characterised by the affective and affectionate relationship between humans.

Thus, it is not surprising that each element of the triad of the French Revolution (1789)—liberty, equality, and fraternity—has seen different developments. While the first two have been widely reflected upon, regulated, defended, and promoted, the last has been successively forgotten. It should also be remembered that the concept of fraternity is not a product of the French Revolution, but has its roots deep in the history of mankind, seen from the perspective of the Western world (Teppa 2012).

## 4. Ecclesial Vision

The contribution we want to make with this research aims to answer to what extent religions—and in this case Catholicism—can make a positive contribution in a world which, due to the growth of globalisation, sees the other as a threat rather than a brother.

### 4.1. Pope Francis

In recent times, the role Pope Francis has played in promoting fraternity must be emphasised. When, on the 4th of February 2019, the Grand Imam of Al-Azhar Ahmad Al-Tayyeb and Pope Francis signed *A Document on Human Fraternity for world peace and living together*, they gave a strong signal of how "religions" can contribute to a more humane and humanising world. In this way, Francis is emerging as a leader with a major global impact (Krames 2015).

In the encyclical Fratelli tutti, the same Pope Francis issues an even sharper challenge when he says: "Fraternity is born not only of a climate of respect for individual liberties, or even of a certain administratively guaranteed equality. Fraternity necessarily calls for something greater, which in turn enhances freedom and equality" (Francis 2020, para. 103).

In a very concrete way, *Fratelli Tutti* intends to explain the growth of being that each person experiences in going out of oneself, in creating a broader web of relationships, until reaching and embodying the sacred duty of effective and fraternal attention towards one's neighbour. This leads to a desire for ever greater good (Di Liso 2021). Fraternity, then, appears to be the highest goal of politics, without which peace and justice are almost impossible. By being understood as a relationship of charity and respect among all members of humanity, beyond all differences that may exist, fraternity thus understood is based on the belief that all humans are "sons" of God, and are his creatures. With this apostolic letter, Pope Francis once again places religious questions at the centre of reflection on a more fair and fraternal world. He emphasises the involvement of religions in social issues (Campdepadrós-Cullell et al. 2021). Since the French Revolution, this has been sought only through politics; he now sees religions as significant allies in achieving an ever clearer awareness of human fraternity. Considering the "something greater" of fraternity, it turns out to be a powerful concept, capable of inspiring humans to build a better world.

We will therefore take a brief look at how Catholicism has come to realise this fact. As the "soul of theology" (DV 21) (Vatican II 1965a), sacred scripture remains an ancient and

at the same time ever-new source for a genuine reflection on fraternity which, intimately united with the concept of "communion" among believers, permeates the two millennia of Christianity's fruitful existence and expands to the whole of social reality. Only an attentive eye can penetrate the web of the relationships generated by an authentic life-giving fraternity and point us in the right direction (de Aviz 2023). In this regard, two New Testament letters are singularly expressive: the First Letter of John (1Jn) and the First Letter of Peter (1Pt).

*4.2. Sacred Scripture*

For the Gospel of John, the love of Jesus is the paradigm of fraternal love (ἀγάπη). It is the love of gratuitousness, which gives birth to communion and bears much fruit (Jn 15:5), it is the love that breaks and dissipates paradigms (Jn 21:15–17), because there is intimacy (Jn 15:15) and giving (Jn 15:17), which awakens the value of the encounter to originate the fraternal bonds of this humanity, and which leaves no room for individualism, but opens itself to mercy characterised by the action of Jesus within this fraternity.

In the First Letter of John, the author provides a theological foundation for the spiritual and human bonds of a fraternity revealed in the event of the incarnation of Christ: what was intangible becomes palpable, for through Christ we are all brothers (1Jn 1–4).

The use of the first-person plural (we) indicates that this relationship of tangible, sensory experience is proper to the community of believers, of brothers and sisters in Christ. The Christ who becomes flesh through the incarnation attracts and generates the Church that makes sensible experience and makes it in community through the use of the we.

> Christian fraternity is born of sonship in Christ (1Jn:3,1). Modern man has not succeeded in building a universal brotherhood on earth, because he seeks a brotherhood without a common centre or origin. They have forgotten that the only way to be brothers is to recognize the origin of the same Father (Documentos do CELAM 2004, para. 241)

According to Johan Konings, John prioritises relationships of fraternity, participation and communion within Christian communities. If they are not properly exercised, the communities cannot enter into dialogue with the world, for it is through fraternal love that the world comes to know them again (Jn 13:35) (Konings 2005, p. 66). In this sense, love is a dynamism which entirely pervades the existence of the Christian community. Paul, eloquent in his "praise of love" (ἀγάπη), appeals to the unity of the communities by showing that they will only live as children of God if they remain in harmony and fraternal union.

Paul emphatically describes love among the virtues as moral behaviour in the Hymn to Love in 1 Cor 13, in which love of neighbour is the full fulfilment of the law (Rom 13:8). Thus, the one who was once a stranger can become a neighbour, not by the ties of consanguinity or affinity, but by the fraternal nature that unites them as brothers (Spicq 1955, p. 370; Klein 1959, pp. 433–34).

The fatherhood of God is extended and expressed in the unity of a concrete community (1Jn 1:3), in which each person united to God through Christ becomes a brother to his neighbour: "so that you may have fellowship with us" (κοινωνίαν ἔχητε μεθ᾽ ἡμῶν) (Smith 2012, pp. 37–38; Burge 1996, p. 55; Stott 2015, p.60). Fraternity is one of the fundamental features of the expressions used by the apostle, a call to brotherly love that moves to compassion and hospitality. What should be characteristic and distinctive for others in order to recognize the Christian should not be the knowledge (γνῶσις) of Christ, but the ἀγάπη (love) that Christians lived, from a sensitive and concrete experience of this reality: the love of the neighbour that calls for the exercise of a universal charity. In other words, the relationship of fraternity must go beyond kinship, to open up to every person, giving rise to a bond of universal love for all those who recognize themselves as brothers and sisters in Christ.

Christian fraternity is the possibility of constituting an indelible stamp on the human being, where the centrality of the mystery of the incarnation of Christ illuminates the relationships between men and women who are invited and confronted by Him to perpetuate

themselves through time in a love that is not imposed. Charity, being a free gift, extends itself unreservedly to those who recognize the importance of bearing in their lives the mark of the Christian, not just because they are admirers of Christ, but because they are his followers by their unique way of acting and thinking, who are able to recognize in the other "Christ himself" (Christianus alter Christus).

Thus, wherever a Christian is, whether in the most different places or in the most different cultures, Christian fraternity is capable of recognizing the other and putting aside one's own differences to open the way to that unity which is the quintessence of being Christian, whose starting point is the love for God, expressed by love for one's neighbour, in the measure of the love one has for oneself.

In 1Pe 1:22, the apostle Peter exhorts his community to live love among the brethren ardently and with a pure heart, in order to experience brotherly love without hypocrisy. This admonition reveals itself as a characteristic note of true fraternity: it is the fruit of a profound ecclesial union and never of segregation. The hypocrite is the one who is fractured or inwardly divided, and therefore unable to open himself to union. The fruits of hypocrisy result in wickedness, lies, and every form of envy and slander (cf. 1Pt 2:1). True religion, therefore, is founded on obedience to the truth, which is manifested in listening and in openness to the other: its consequence is action in the form of a fraternal love that excludes any form of division.

Looking at this Letter written in the second half of the first century AD and addressed to "those who live as strangers", that is, those who live away from home (cf. 1Pt 1:17), it is relevant because it speaks to those who have left in search of better conditions, because they were slaves or lived far from their homeland. Indeed, the Christian seeks his definitive homeland, the heavenly Jerusalem.

They were men and women who had been cut off from their families and cut off from their friends, who had cast off their roots, and who found themselves in places and regions which were not able to offer the same hospitality and welcome as was found in the family hearth. These Christians, being foreigners, were exposed to humiliation and slander. The author of the Letter consolidates the stable element in the life of an immigrant as the basis of Christianity: true brotherhood. This expresses unity as the distinctive sign of a true community, where welcome and love mark the face of the Church, not only as a "house of God" or assembly, but as ἀδελφότης: a place of those who love one another with a fraternal and unfeigned love.

In this text, Christianity is revealed as an expression of fraternal love, described by the adjective ardent, rationalistic or cold, but enthusiastic and ardent, capable of scorching even the most indifferent to the faith. It is not unintentional that this Letter was written in Rome, demonstrating the universality and imperative of Christianity, making itself known as a Church capable of welcoming, loving and communicating a universal love.

The author's admonition to the brothers who compose this fraternity takes up the concept that they "be all of one mind, compassionate, full of brotherly love, merciful and humble in spirit" (1Pt 3:8), that is, not some, but all. What seems an arduous task becomes a distinguishing mark of the Church. This is *fraternity* only to the extent that the brothers are able to look at each other, regardless of the other's social status, language or culture, but as one who is part of the same family.

The Christian novelty that we find in this Letter is found in the universal call for a love that is binding, no longer by the ties of consanguinity—typical of members of the same family or clan—nor by an egalitarian reciprocity that characterises the φιλία, but by a universal and gratuitous love that unites, regardless of cultural ties. Consideration of the dignity of the human person thus establishes a universal bond in Christ, in which we are all brothers and sisters in one Father. The universality of this love is found in the term ἀδελφός, which underlies the concept of community as φιλαδελφία (Malcolm 2019).

*4.3. The Originality of Christianity*

Fraternity is what amalgamates the visible reality of the Church. St Ignatius of Antioch (+107), in his famous adage about the Roman Church presiding in love (προσκαθημένη τῆς ἀγάπης) (d'Antioche 1958, pp. 124–25; Schoedel 1985, p. 166), affirms the primacy of Rome as complementary to the equally relevant aspect, namely its nature and its *modus operandi* to exercise and establish itself as a visible sign of an invisible reality, in dialogue and fraternal welcome, in service and not authoritarianism, in listening to the most diverse segments and ecclesial structures, and as the exercise of a synodality which, through a dynamic and conscious action of all its members, is capable of promoting the growth of a fraternal κοινωνία, where differences are not an obstacle to sharing the same bread: "Since there is one bread, we, though many, are one body, since we all partake of that one bread" (1 Cor 10:17).

The originality of Christian fraternity is the ability to love others without distinction, including enemies; but not naively.[2]

This is the teaching of his words: "Bless those who curse you and pray for your enemies, and fast for those who persecute you. What merit is there in loving those who love you? Do not the pagans do the same? As for you, love those who hate you, and you will have no more enemies" (Rordorf and Tuilier 1978, pp. 143–45).

A Church that claims to live in communion but does not walk and practise the truth is a liar (1Jn 1:6–7); however, those brothers who are not in communion defraud Christianity. If it does not bear witness to brotherly love, it is deficient: "Little children, let us not love in word or tongue, but in action and in truth" (1 Jn 3:18).

Fraternity, however, builds not only the Church but also civil society. For Raniero Cantalamessa, social sentiment "was born in the soil irrigated by the Gospel and that in the first centuries and throughout the Middle Ages, the means par excellence for acting in the social sphere and reaching out to the poor was almsgiving" (Cantalamessa 2022).

Almsgiving, recognised as a great virtue in the Judaeo–Christian mentality, still retains its relevance and its contemporaneity; however, it cannot be the only current and ordinary way of acting in a social fraternity, as Pope Francis very well defined in his encyclical *Fratelli Tutti*. Almsgiving does not safeguard the dignity of the needy, because it perpetuates their state of dependence, since almsgiving, as an expression of human solidarity, must not nourish a state of dependence which would hinder the capacity for self-assertion and self-promotion.

A social fraternity positively impels society (Bourdin 2022, pp. 137–39), organisations and institutions, as well as their leaders, to effective practices of public policies that are capable of undertaking structural processes which reduce the indecent abyss of social inequality.

Moreover, it is capable of safeguarding a trend that, if not corrected, causes the very societal dehumanisation and everything that concerns communion and relationships (Breen 2010; Melé and Naughton 2011).

According to the Christian humanist Lactantius (+330), "the strongest bond that unites men is humanity" (Lactancius 1973, p. 670). It strengthens bonds, enabling every human being to experience that sense of belonging and welcome which is proper to authentic fraternity.

It is in the original core of humanity that God necessarily makes himself present, for in creating man for the covenant (ברית), he makes every human being recognise the other as brother, in order to generate a universal brotherhood in which all are recognised as the image and resemblance of God himself (Gen 1:26–27). But if all are ἀδελφοί, what makes men so dehumanised as to hurt the one who bears my own image? If they want to deserve to be recognised as men and children of God in Christ, for Lactantius they must show their humanity, which means to love man because he has the same nature that we have in common (Lactancius 1973, pp. 666–67).

In the wake of this thought, the philosopher Emmanuel Lévinas presents fraternity as an ontological element, as what constitutes reality and the human being: a being open to

the other: "The human self is situated in fraternity: the fact that all men are brothers is not added to man as a moral achievement, but constitutes his ipseity" (Lévinas 2008, p. 278).

It is observed that the Second Vatican Council (Vatican II 1965b, 1965c, 1965d, 1965e), clearly recognized the value of fraternity in the most diverse human realities, which must always be preserved. According to Charles André Bernard, "this theoretical and practical recognition of the Church underlines its character as a specialist in humanity" (Bernard 1999, p. 19) and fraternity.

Based on this reality, in their encyclical letters, Popes Benedict XVI and Pope Francis insist that fraternity (Benedict XVI 2023, para. 19; Francis 2020, para. 12) is not a Herculean work aimed at achieving a morality whose force is dominant. Fraternity is not imposed, otherwise it would be the fruit of violence. The simple intellect, on the other hand, can see equality and right among men, but it cannot find fraternity. For this intellect, fraternity has its origin in a transcendent vocation, in a God who is Father because he first loved us, teaching us through his Son what fraternal charity is.

The model of relationships between persons, in the thinking of the magisterium in the light of sacred scripture, must have as its objective the awakening to the awareness of a social friendship, as announced in 1 John, which is proposed to be mirrored in the triad of friendly relationships, as an invitation to enter into the mysterium amoris which never closes, but remains open to the plurality of persons through love, which seeks to build a universal fraternity. This will grow to the extent that one offers what is best for the other, simply by loving them and being who they are, as a true greater good.

The relevance of reflecting on the importance of fraternity in the construction of societies is supported by the global challenge of adopting a response to the public agenda for a collective political and social effort that safeguards shared responsibility; Christianity is rediscovered as a facilitator of opportunities that aim to overcome economic and social ties to rescue the dignity of every human being (Losh 2022). After all, the theological and humanist core lies in an *ethos* of universal fraternity that leads us to work towards a culture of hospitality and inclusion.

From this universal fraternity, the pontifical teaching has always sought to dialogue with and foster the anthropological–social core of modernity. Since human beings are indebted to one another, and in order for humanity to be reborn with all faces, all hands and all voices, these must be free from the borders that they themselves construct (Francis 2020, para. 35). Freedom, equality and fraternity have always been values of the Christian tradition, and after the French Revolution they became pillars of modern culture. This historic event praised these three words as the emblem of a new stage in universal history, but just forgot to look at and deepen their origins.

**Author Contributions:** Writing—original draft preparation, D.A.d.A. and L.M.F.R.; writing—review and editing, D.A.d.A. and L.M.F.R. All authors have read and agreed to the published version of the manuscript.

**Funding:** This research received no external funding.

**Data Availability Statement:** Not applicable.

**Conflicts of Interest:** The authors declare no conflict of interest.

## Notes

[1] Liogier (2012), Chapter Une question de style.

[2] Cf. Mt 5, 44s; Lc 6, 27s; 6, 32s.

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
