# Peer review of "Christianity and Fraternal Globalisation"

_religions, doi:10.3390/rel14101234_

Round 1
Reviewer 1 Report
The author has an interesting purpose in describing the "Something greater" (line 31) that Pope Francis contributes to the role of fraternity, but does not clearly articulate or argue this in the paper.
The author needs to make better connections throughout the essay. For example, in the section on contextualization, he attempts to connect "myth" and "religion" with globalization but fails to describe the meaning of either.
The section on Political motivations" does not get connected to the "something greater" nor to what follows.
The section on Pope Francis has some good insights, but lacks cohesiveness and logical connections between the paragraphs.
Section 4.2 on 1 Peter - what is the source for his contextualization in paragraph 1 & 2? This section needs to include more of the content of 1 Peter to make the arguments.
In 4.3 he goes to "Patristics" but only after that treats of 1John - this is out of order. He fails to demonstrate how the Incarnation is central to the thought of 1 John he merely affirms it. (Lines 199-207 is one sentence!!!)
Line 4.4 - what is "This"?
Lines 243-245 - is this a direct quote? If so, indicate this. Lines 251-254 - another direct quote?
Lines 266-267 - What "trend"
The conclusion introduces new material!
This essay has potential, but needs much work in language and expression, and especially in organizing the argument so that it flows with concrete connections throughout.
The lack of quotation marks and indication of author in direct quotes needs attention!
Line 57 - "in a facilitating role"?
Line 132 - final clause hanging
Lines 179-181 - difficult to understand
Line 191 - what "Charter"
English needs refining.
Author Response
Dear Reviewer
We are very grateful for your excellent comments, which have helped us a lot.
You wrote that we had an interesting goal in describing the "Something greater" that Pope Francis contributes to the role of fraternity, but we did not argue it clearly in the text. This aspect has been improved. This can be seen, for example, in the lines @ 179 to 190.
We have established a better relationship throughout the various aspects addressed in the text. We have also described more explicitly the concepts of "myth" and "religion", to understand their connection with globalization.
We have also extensively redrafted sections 4.1, 4.2 and 4.3 to respond to the Reviewer's comments. The text has now been placed in biblical order, respecting the historical order of the books cited in this article. (lines @ 198-292). The patristic item has been deleted and put back in an organic and coherent way within the text (line @ 294-303).
The thought of how the Incarnation is central to the thought of 1 John, is now demonstrated and grounded biblically. (lines @ 207-211).
When you ask us (Line 4.4 - what is "This"?), we have suppressed and put more clearly the thought. (lines @ 304-305)
The remark you made about lines 243-245, if they were a citation, we now quote correctly and with the corresponding quotation marks (lines @ 295-296 and 306-309).
When you ask us "What tendency", and although the lines do not match, but we believe you are referring to the lines "blood ties kinship to open up to others, every person giving rise to a bond of universal love for all those who recognize themselves as brothers and sisters in Christ." ( @ lines 235-237).
We remove the title conclusion. In fact, as the text developed, it no longer made sense to keep it. We should have taken it out at the very first submission, not least because it is a non-mandatory title.
Once again, we reiterate our gratitude.
Reviewer 2 Report
This submission in structure, style and tone more akin to a homily or sermon than a fully-formed academic paper. The preference for 'cf' references suggests that close engagement with the critical context is not the author's priority. The sequence of short - sometimes scarcely more than a single sentence - paragraphs reinforce the impression of a collation since they privilege assertion over critical analysis. It is not clear to what research debate this essay responds; rather it appears to be quasi-scholarly reflections on recent papal encyclicals.
This submission in structure, style and tone more akin to a homily or sermon than a fully-formed academic paper. The preference for 'cf' references suggests that close engagement with the critical context is not the author's priority. The sequence of short - sometimes scarcely more than a single sentence - paragraphs reinforce the impression of a collation since they privilege assertion over critical analysis. It is not clear to what research debate this essay responds; rather it appears to be quasi-scholarly reflections on recent papal encyclicals.
Author Response
Dear Reviewer
We have taken a close look at your comments, which we thank you very much. You have helped us greatly to improve our work.
We have improved the quality of the text, both in content and in the reflection it presents, to try to respond to this suggestion, which we consider important.
We are very grateful.
Reviewer 3 Report
Christianity and fraternal globalization
The work is an interesting proposal on the relevance of fraternity to improve relations between people and combat the negative effects of globalisation. It reviews the term fraternity as an element of the triad proposed in the French Revolution together with liberty and equality. In any case, the work underlines the origin of this fraternity as a basic element of the message of Jesus Christ himself, as an element of Christian identity. In order to justify it, elements from both Sacred Scripture and the Fathers of the Church appear. It also adds explicit references to fraternity in the writings of Popes Benedict XVI and Francisco.
The article presented reinforces the need to value fraternity, born in Christianity and promoted by the Revolution of 1789, as an essential commitment for today's society where division and individualism have contaminated relations between people.
This proposal to promote fraternity is an original and interesting initiative to improve the world, for which the author or authors of the work are to be congratulated.
Based on this assessment, I consider it necessary for the author(s) to reflect on the following points in order to give more depth to the work presented:
1. Necessary modification. In the Christian context, it is necessary to relate fraternity to the concepts of love and mercy. For this reason, other references would be necessary, such as the texts of the Gospel of John and Paul's first letter to the Corinthians.
2. Relevant proposal. Within the framework of relations between cultures and the development of globalisation, the proposals of Amartya Sen, especially in his work Identity and Violence (2015), are widely recognised in the world and add value and significance to the article.
3. Relevant proposal. At present and within the framework of the globality proposed in the article, Professor Flecha's book The Dialogic Society (2022) is a necessary work to justify the path that has brought us to a society that, fleeing from dialogue, has fostered violence in human relations because it has considered authors who have justified violence or the lack of criteria of truth and authenticity as referents in thought.
4. Proposal for reflection by the author(s). A leap is made from Sacred Scripture to the last Popes, however, within the Catholic framework, references could be made to the Second Vatican Council (Lumen Gentium, 1964; Gaudium et Spes, 1965; Nostra Aetate, 1965; Ad Gentes, 1965).
5. Proposal for reflection by the author(s). Since Christianity is presented as a promoter of fraternity, it would be interesting to reflect on how this vital attitude is promoted in relations with other religions. For this purpose, the article published in this journal under the title Interreligious Dialogue Groups Enabling Human Agency (2021).
Once again, I would like to congratulate the author(s) and encourage research to improve human relations and work for a freer, more egalitarian and fraternal world.
Author Response
Dear Reviewer
We are grateful for your very relevant suggestions. They helped us to see aspects we had not noticed and to broaden our own thinking.
Based on each of your suggestions, we have made the following changes:
1 - The mercy, although this term is not used ipsis litteris by the author of the Fourth Gospel, is found in the semantic force of the term, to the modus operandi of Jesus characterized by his divine action, an element that is not absent in the Gospel of John, whose mercy is related to love. On this basis, two paragraphs were developed with references in the Gospel of John (lines @ 199-204; @ 216-222) and the 1st Letter of St. Paul to the Corinthians (lines @ 220-226).
2 - We consider Amartya Sem's proposals in the framework of relations between cultures and the development of globalization. It was indeed a suggestion that added much value to our text. (Lines @ 97-105)
3 - We have also inserted some reflections by Ramon Flecha, mainly from his book The Dialogic Society. We have found it to be an important work to justify the path that has led us to a society that, eschewing dialog, has fostered violence in human relations by considering authors who have justified violence or the lack of criteria of truth and authenticity as referents of thought.
The thought of Prof. Ramon Flecha, as well as that of Amartya Sem, have introduced great value into the text (Lines @ 107-105).
4 - A paragraph (lines @ 346-350) referring to the Second Vatican Council has been contemplated. Indeed, it is important to mention it.
5 - Finally, the article entitled Interreligious Dialogue Groups Enabling Human Agency (2021) has been considered in this text.
Thank you for your important recommendations.
Round 2
Reviewer 2 Report
The author has responded to the review with some care and consideration and has made systematic changes to their paper. I remain concerned that the sum is a somewhat slight contribution to the scholarly literature but in its own terms it is now well-articulated and adequately supported with references. I would still wish to see a more convincing case made in the opening for the wider research significance of the topic and in the conclusion for the value of this contribution.
Author Response
Dear Reviewer
We have looked very carefully at your suggestions, which have greatly helped to improve our work.
In the introduction to the paper, we explained why we chose this topic. In the conclusion, we have endeavoured to highlight the contribution that this work makes to the theological reflection that is taking place on sustainable and fraternal development, understood in the broadest sense.
We remain at your disposal,
The authors.